# Structural Home Defects Are the Leading Cause of Mold in Buildings: The Housing and Health Service Experience

**DOI:** 10.3390/ijerph192416692

**Published:** 2022-12-12

**Authors:** Rachel Felipo, Denis Charpin

**Affiliations:** 1French Clean Air Association, 59120 Lille, France; 2Department of Pulmonology and Allergy, Aix-Marseille University, 13005 Marseille, France; 3Marseille Innovation—Hotel Technologique, 45 Rue Joliot-Curie, 13013 Marseille, France

**Keywords:** housing, unhealthy housing, unsanitary housing, mold, rehabilitation, public health

## Abstract

The study aimed to evaluate the contribution of building problems to mold proliferation in dwellings. We investigated 503 dwellings of patients suffering from respiratory diseases, whose attending physicians had requested a home inspection if the area of mold was equal to at least one square meter. After careful visual evaluation and basic environmental measurements performed by a trained technician, environmental issues were classified into building defects, accidental water damage, and condensation. Data analysis demonstrated that building defects were the pre-eminent cause of mold proliferation. Among the building defects, water infiltration through leaks in roofs or walls was the leading cause. These results highlight the need for health professionals managing patients with respiratory diseases to be able to request a home inspection, and for city health authorities to commission professionals who can focus on building problems and find ways to address them.

## 1. Introduction

Mold proliferation has been identified in around 20% of dwellings in Europe [1] and the United States [2]. It has important short- and long-term health impacts [3].

Several governmental and nongovernmental organizations around the world have issued guidelines on the health impacts of indoor mold [3,4,5]. Home assessments of buildings with mold have highlighted many different risk factors, such as building characteristics, building defects, family structure, and living habits. However, the relative importance of these risk factors has not been evaluated.

In practical terms, when a home inspection is performed, the technician should be able to identify the leading cause of mold proliferation.

This paper aimed to identify the leading causes of mold proliferation, following home visits to buildings with mold problems. This issue is crucial because it affects the way in which the building will be managed: either through remediation undertaken by professionals in the case of structural defects, or simple adjustments or counseling in the case of inappropriate occupant behavior.

## 2. Material and Methods

### 2.1. Studied Buildings

Five hundred and three dwellings with mold were studied. A dwelling was considered to have mold if the sum of the surfaces with mold in the living areas (including walls, ceiling, furniture, curtains, clothes, and shoes) was equal to at least 1 square meter (stage 1 or above of the scale adopted by the New York City Department of Health and Mental Hygiene [4]). The initial assessment of the moldy surface was performed by a technician who had carried out hundreds of home visits over many years. These buildings were inhabited by patients whose attending pulmonologists considered that their living conditions could worsen their respiratory condition. Indeed, twenty years ago we established a service to allow pulmonologists to order such environmental inspections [6]. All the buildings were located in Marseille, in southern France. Most of the dwellings were located in high-rise buildings built in the late 1960s and early 1970s, using cinder blocks. All home visits were performed by the same technician from 2013 to 2019.

### 2.2. Methods

The technician first performed a careful visual inspection of the dwelling. In each room, they located possible sources of air pollution by checking the condition of the joinery and heaters, maintenance of combustion appliances, looking for spots of mold, and checking the ventilation. They also had a discussion with the tenant concerning the living habits of the family and, in particular, the use of cleaning products.

The technician then completed a questionnaire regarding the above items, and also the number of occupants.

Then, various types of equipment were used in the living room and the patient’s bedroom, as follows:-A Qtrack to measure ambient temperature and relative humidity (Q-Trak Indoor Air Quality Monitor 7575, TSI Inc., Shoreview, MN, USA);-A mural hygrometer (Protimeter survey master (Protimeter plc, Marlow, UK));-A light paper sheet and a smoke canister to check the efficiency of mechanical ventilation (Smoke pen Bjornax AB, Nora, Sweden).

At the end of the visit, environmental issues were classified into three different groups:-Building defects: rising dampness, water infiltration through leaks in the roof and/or external walls, thermal bridge;-Accidental water damage resulting from a sudden disruption of the water pipes;-Vapor condensation on walls and/or ceiling. Measurement of wall humidity allowed the cause of mold development to be identified as either vapor condensation (low wall humidity) or a water leak (very high wall humidity).

## 3. Results

The study group included 67 dwellings without structural defects, and 436 with at least one defect. The mean number of occupants per dwelling was equal to 6.25, although the mean figure for this area is 2.5 [7].

Figure 1 shows the distribution of the causes of mold infestation. Almost half were related to building problems.

Figure 2 lists the main building problems leading to mold proliferation: water infiltration, damp rise, and thermal bridge. As shown, among the buildings where mold proliferation was identified, 72% were caused by water infiltration, 21% were caused by damp rise, and the last 7% were caused by thermal bridges. Therefore, water infiltration was the major contributor to mold proliferation; the number of buildings where mold proliferation was caused by water infiltration was more than ten times the number of houses where mold proliferation was caused by thermal bridges. Figure 3 displays the ventilation status. The latter was functional in less than a quarter of the dwellings.

Figure 3 shows the status of the ventilation, which was either functional, i.e., arranged according to current regulations and working in less than a quarter of the dwellings; faulty, i.e., including missing air extraction vents, insufficient airflow, or missing decalibration of doors in 37% of dwellings; or absent, when there was no ventilation system, in 40% of dwellings.

## 4. Discussion

This study has strengths and limitations. Its strengths are the large sample size and the standardization of the visits performed by a single trained technician. Nevalinen et al., have demonstrated that occupants missed signs of current or previous mold problems in 15% of houses surveyed [8]. In addition, in a study of 291 Finnish houses, Pirvinen found that two-thirds of mold problems could be identified by nondestructive methods, such as careful visual inspection [9]. The limitations are the limited number of items relative to the dwelling and its usage, and the fact that most dwellings were located in high-rise buildings, which impairs the generalization of our findings to other types of dwellings.

Existing literature has highlighted many different risk factors for mold development: age of the building [10], number of occupants [11], private houses compared to apartments [11,12], and, for the latter, living on the first floor compared to higher levels, low ventilation rate [11], and low socioeconomic status [10]. In addition to those factors, other papers have stressed the relevance of the structural characteristics of the buildings to mold development: types of foundations [12,13], water damage [14,15,16,17], lack of basement [17], and faulty wall insulation [18]. Finally, mold proliferation can result from, or be exacerbated by, the occupants’ behavior such as blocking ventilation, or producing excessive water vapor [18,19].

From the abovementioned results, it is clear that building problems are the number one cause of mold proliferation. Indeed, in addition to the “building problems” category, such problems also feature in the “condensation” category when condensation is caused by thermal bridges, and a lack of proper ventilation systems.

To the best of our knowledge, this is the first demonstration of the pre-eminence of building problems as the main causes of mold proliferation. Inappropriate building usage is shown to be merely a contributory factor. Therefore, city health authorities and home inspections should first focus on possible building defects. Similarly, health professionals who manage patients suffering from respiratory diseases that may be related to mold proliferation should advise these patients to ask for a home inspection, because it is unlikely that the common advice they offer, concerning ventilation and cleaning moldy surfaces, will resolve the issue.

## 5. Conclusions

In this study, we aimed to identify the causes of mold proliferation in dwellings. In inner-city high-rise buildings, building problems are the main cause of mold proliferation. This result is likely to apply to this type of building in other locations, but not necessarily to other types of housing in other climates. These findings point to the need for home inspections by professionals so that the defects identified can be correctly addressed.

## Figures and Tables

**Figure 1 ijerph-19-16692-f001:**
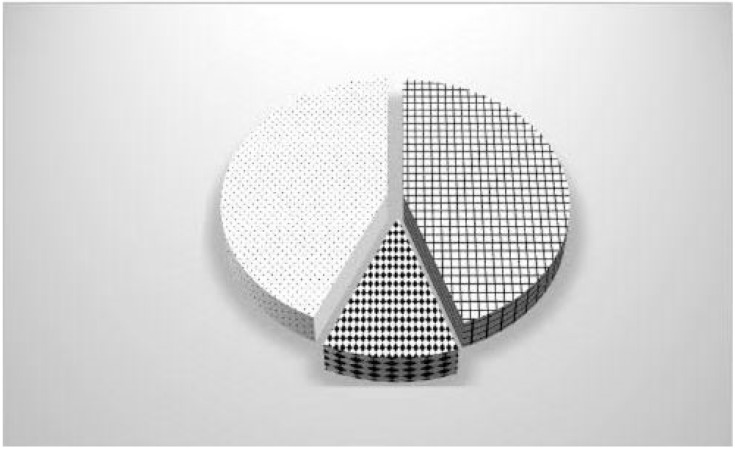
Distribution of the causes of mold infestation according to building problems 45% 
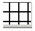
 water damage 12% 

 or condensation 43% 
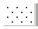
.

**Figure 2 ijerph-19-16692-f002:**
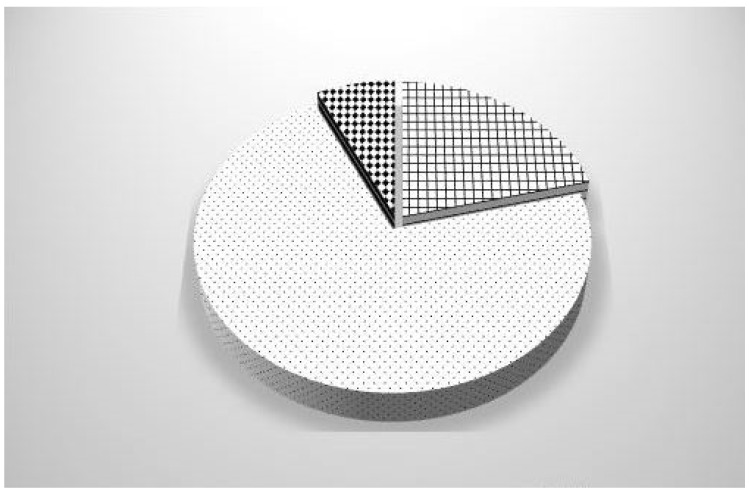
Distribution of building problems: damp rise 21% 

, water infiltration 72% 
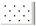
 or thermal bridge 7% 

.

**Figure 3 ijerph-19-16692-f003:**
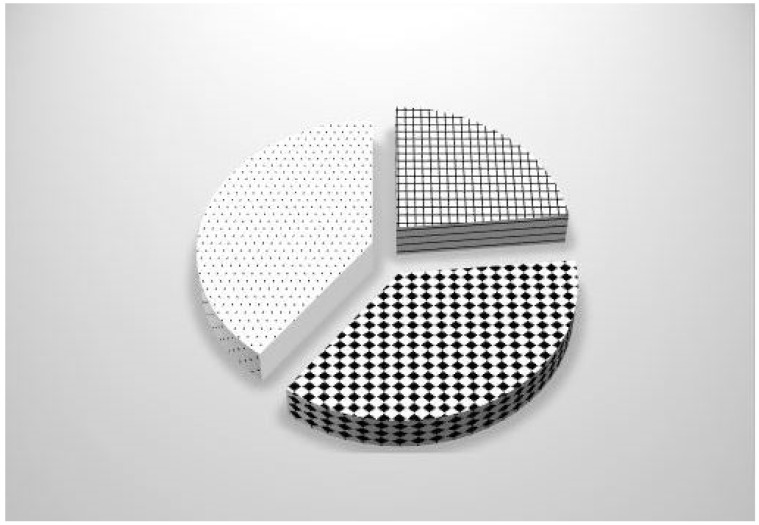
Distribution of ventilation status in the dwellings: functional 23% 

, faulty 37% 

 or absent 40% 
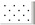
.

## Data Availability

Not applicable.

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
