# Peer review of "Structural Home Defects Are the Leading Cause of Mold in Buildings: The Housing and Health Service Experience"

_ijerph, 2022, doi:10.3390/ijerph192416692_

Round 1
Reviewer 1 Report
This is a great short report on a very interesting topic. The paper is concise and well written. One minor thing for the authors to consider including is a detailed description of the categorizations used in the three figures.
Reviewer 2 Report
This study investigated the likely source of mould proliferation in buildings with at least 1 square metre of mould, by using building inspection to examine 344 dwellings in southern France.
This study addresses an important knowledge gap, as very few studies investigate the likely moisture source of mould proliferation. This is an interesting topic of investigation, and the large sample size is clearly a strength of the study. However, there are a number of gaps in the description of the study methods which make it difficult to ascertain how likely the results from this study would translate to other building types in other climates.
Comments/questions
1. Who made the initial assessment that a house had a 1 m2 area (or more) of mould – was this the occupants or an inspector?
2. To what extent was the ‘technician’ trained in identifying moisture issues? Many studies employ specialist building assessors, who may still struggle to identify water source/moisture problems without employing destructive sampling (estimated by Pirinen 2006 to be a third of houses in Finland (WHO 2009)). How skilled must the technician be? Could using a specialist building engineer or destructive sampling given different results?
3. A clearer explanation of what criteria were used to make decisions around whether mould was caused by condensation or leaks would be helpful.
4. What variety of buildings and building materials were covered by the study, and to what extent do the authors believe the results found in this study are applicable to other building types/ climates/other areas of the world? A summary/description of the types of dwellings included would be helpful (construction year, single story, standalone, apartments, predominant building materials etc).
5. Was location of the areas of mould and water damage in the dwelling also documented? If so it would be good to include in the results, as previous studies (eg Pekkanen et al 2007 Eur Respir J 29:509-515) suggest that location may be important for adverse health effects.
6. Did mouldy surfaces over 1m2 include mould only on walls and ceilings? As no mention is made of walls on other surfaces of the dwelling, such as on windows/curtains etc.
7. It is unclear from the current method description what the careful visual inspection entailed, and what was being looked for during this, and also whether all of the techniques listed were employed in each house, and in each room and on what materials?
8. What is meant by accidental water damage? And how does this differ from a leak in definition in this study (eg is it a shorter timeframe of occurrence?).
9. While other studies specifically looking at causes of moisture issues in homes with of 1m2 of mould are limited, it might be helpful to discuss the literature that does exist already on the sources of moisture issues in homes, such as Nevalainen et al. Indoor Air 1998 S4:45-49, Pirinen 2006 (mentioned in WHO 2009) etc as a comparison
1. Limitations of the study need to be mentioned.
1. Conclusion is too broad – this needs to be more specific eg for this study in these types of buildings/dwellings this was the case.
Minor typos
L22, no need for a comma between Europe and United States
L27 – full stop in middle of a sentence. Sentence would benefit from rewording to improve flow.
Reviewer 3 Report
This manuscript described an investigation on mold proliferation in dwellings. The purpose of this study is very clear and the sample size is big enough. However, the methods part, especially the classification of environmental issues, is doubtful and needs to be validated. In addition, the results part is too short. It only contains three figures; more data and explanations are expected. Last but not least, the English writing and usage of punctuation also need to be improved. The following detailed comments could help the authors to revise the report:
Introduction
1. Please check the punctuation. For example, in the first sentence, the “,” should be removed and a “.” should be added at the end of it.
2. Please add the impact of mold proliferation in this part. Why it worth to be studied?
3. Paragraph 2, what do the authors mean by “risk factors”?
Material and Methods
4. Please heck punctuation. All the 《》 should be changed to “”; “)” is missing in line 39.
5. What is ‘stage 1 or up of the Bureau of Hygiene of New York City’? and what is “Bureau of Hygiene of New York City”? Do you mean ‘New York City Department of Health and Mental Hygiene’?
6. Line 47, lots of apparatus can be used, but were they really used? If so , please use ‘were’ instead of ‘can’. Besides, you don’t need to separate this line into a separate paragraph.
7. Line 48, what do you mean by ‘hygrometry’? Do you mean “relative humidity”?
8. Line 54, how do you select these groups? From the reviewer’s perspective, there is overlap between these groups, for example thermal bridge will cause ‘vapour condensation on wall and/or ceiling’. This classification is the foundation of this study, so it needs to be explained and validated carefully. Besides, some background information about the environmental issues should be provided in the introduction part. Are there any other studies that investigated these issues before? Perhaps based on previous studies, the authors could find a better classification of the issues.
Results
9. Please provide legends on the figures, and add the percentages on the corresponding areas on the figures.
10. Please add more description, explanation for the figures.
11. In the methods parts, it was mentioned that temperature, CO2, ect. Were measured, what are the measurement results? If the authors don’t want to show them in this report, then do not mention them in the methods part. The method section and the result section should correspond to each other.
Discussion
12. Line 79, please delete the space before ‘Apart…’.
13. Line 80-81, ‘type of foundations’ and ‘lake of basement’ are not structural defects.
14. Lines 85-87, please rephrase this sentence.
15. Line 95, please add a ‘.’.
Conclusion
16. Please check punctuation.
17. Please add more content in this section, for example, restate the research question, summarize the finding and mention the implication...
18. Reference 16 was not cited in this paper.
Round 2
Reviewer 3 Report
Thank the authors for patiently reviewing the comments, most of the comments have been addressed properly. I think the revised version is now in much better shape. However, the results part is still too short. To make it clearer and easier to be followed by the readers, more descriptions of the figures are suggested be added. For example, for figure 2, you could introduce what are the main building problems that led to mold proliferation, and provide the percentages of these problems, then mention which one is the most popular problem and which one is the least. An example could be “Figure 2 lists the main building problems leading to mold proliferation: water infiltration, dam rise, and thermal bridge. As is shown, among the buildings where mold proliferation was identified, 72% of them were caused by water infiltration, 21% of them were caused by damp rise, and the last 7% were caused by the thermal bridge. Therefore, water infiltration was the major contributor to mold proliferation, the number of buildings where mold proliferation was caused by water infiltration was more than ten times as the number of houses where mold proliferation was caused by thermal bridges.” For the other figures, similar descriptions are expected. Besides, the sentence “Figure 3 shows the status of the ventilation which was functional in less than of quarter of the dwellings” has grammar issues. Last but not least, please highlight all the changes.
Author Response
Dear colleague,
Thank you again for your help in improving this manuscript.
In this new version, we have added
-in the "Methods" section, in lines 68-69: "and checking the ventilation" and, in line 78 "A light paper sheet"
-in the "Results" section, in lines 95-102, comments on figure 2 and, in lines 104-107, comments on figure 3. We have used the word "decalibration" to mean cutting the lower part of the door to facilitate air movements.
We hope the paper is now suitable for publication
Best regards
